# A Blended Vitamin Supplement Improves Spatial Cognitive and Short-Term Memory in Aged Mice

**DOI:** 10.3390/ijms25052804

**Published:** 2024-02-28

**Authors:** Koji Fukui, Fukka You, Yugo Kato, Shuya Yuzawa, Ayuta Kishimoto, Takuma Hara, Yuki Kanome, Yoshiaki Harakawa, Toshikazu Yoshikawa

**Affiliations:** 1Molecular Cell Biology Laboratory, Department of Bioscience and Engineering, College of System Engineering and Science, Shibaura Institute of Technology, 307 Fukasaku, Minuma-ku, Saitama 337-8570, Japanmf22137@shibaura-it.ac.jp (S.Y.); mf22100@shibaura-it.ac.jp (T.H.);; 2Division of Anti-Oxidant Research, Life Science Research Center, Gifu University, 1-1 Yanagito, Gifu 501-1194, Japanharakawa@antioxidantres.jp (Y.H.); 3Antioxidant Research, Louis Pasteur Center for Medical Research, 103-5 Tanakamonzen-cho, Sakyo-ku, Kyoto 606-8225, Japan; toshi@yoshikawalab.jp; 4Kyoto Prefectural University of Medicine, Kajii-cho, Kawaramachi-Hirokoji, Kamigyo-ku, Kyoto 602-8566, Japan

**Keywords:** mixed antioxidants, cognition, coordination ability, short-term memory, aged mice, training effect

## Abstract

Although many types of antioxidant supplements are available, the effect is greater if multiple types are taken simultaneously rather than one type. However, it is difficult to know which type and how much to take, as it is possible to take too many of some vitamins. As it is difficult for general consumers to make this choice, it is important to provide information based on scientific evidence. This study investigated the various effects of continuous administration of a blended supplement to aging mice. In 18-month-old C57BL/6 mice given a blended supplement ad libitum for 1 month, spatial cognition and short-term memory in the Morris water maze and Y-maze improved compared with the normal aged mice (spontaneous alternative ratio, normal aged mice, 49.5%, supplement-treated mice, 68.67%, *p* < 0.01). No significant differences in brain levels of secreted neurotrophic factors, such as nerve growth factor and brain-derived neurotrophic factor, were observed between these two groups. In treadmill durability tests before and after administration, the rate of increase in running distance after administration was significantly higher than that of the untreated group (increase rate, normal aged mice, 91.17%, supplement-treated aged mice, 111.4%, *p* < 0.04). However, training had no reinforcing effect, and post-mortem serum tests showed a significant decrease in aspartate aminotransferase, alanine aminotransferase, and total cholesterol values. These results suggest continuous intake of a blended supplement may improve cognitive function and suppress age-related muscle decline.

## 1. Introduction

As the population ages, the number of people suffering from chronic diseases such as Alzheimer’s disease and diabetes is increasing [1,2]. It is easy to predict that the number of patients with these diseases will continue to increase in the future, and as a result, the medical costs for treatment will increase enormously, putting pressure on public finances [3]. Although various drugs are currently available for these diseases, no clear treatments or drugs are available for many other age-related diseases. Significant basic and applied research focuses on the development of new drugs for these conditions, with large investments in human resources and research funds. Therefore, although it is important to develop therapeutic drugs to prevent these diseases, it is also necessary to take preventive measures as early as possible before age-related diseases develop. 

Oxidative stress is involved in many aging-related diseases [4]. The body takes in oxygen through breathing, but some of this oxygen becomes active and attacks living tissues [5]. As a result, oxidation products gradually accumulate in the body, increasing the risk of developing or exacerbating age-related diseases [6,7]. Furthermore, Harman’s free radical aging theory postulates that the accumulation of oxidation products itself accelerates aging [8]. Therefore, to prevent or delay the various functional declines associated with aging, it is important to prevent oxidation in the body continually [4]. 

To prevent oxidation, the human body is equipped with an antioxidant system that detoxifies the products of oxidative stress [9]. Redox balance in the body is maintained through the action of antioxidant enzymes such as superoxide dismutase, catalase, and glutathione peroxidase [10,11]. These enzymes alone are not sufficient, so antioxidants must be taken in through the diet to provide adequate protection from oxidation. As we age, the redox balance gradually shifts toward oxidation [12]. This may be due to a variety of factors, including the weakening of in vivo antioxidant defense mechanisms due to aging and exposure to external stress [13]. Therefore, it is difficult to maintain the redox balance associated with aging simply by ingesting antioxidants from the diet. 

Many people take supplements to compensate for inadequate dietary antioxidant intake. These products contain a variety of antioxidants, polyphenols, polyunsaturated fatty acids, and plant extracts. In Japan, the demand for such products is increasing yearly, with a market value approaching 1 trillion yen [14]. However, because so many types of supplements are available, it is very difficult for general consumers to know which to take and in what amounts for what purpose. Additionally, there are excesses of some vitamins. Uncertainty regarding supplements causes problems for sellers and consumers, and since they must be purchased, suppliers are required to provide accurate information based on scientific evidence. 

Blended supplements are one way to solve these problems [15]. As the dosage is determined in advance, consumers do not need to worry about overdosing. One of these blended supplements is Twendee X (TwX), which contains eight types of active ingredients and has been patented in Japan. Furthermore, it was the first supplement to be certified as “Grade A” by the Japan Society for Dementia Prevention as being effective in preventing dementia. This blended supplement has also been clinically tested [16] and has more evidence than others [17,18,19]. However, there is still not enough scientific evidence. There is little scientific evidence for this blended supplement specifically for motor function. In this study, we used TwX to investigate various effects of blended supplement intake in aged mice, including motor function.

## 2. Results

### 2.1. TwX Does Not Affect Body Weight in Aged Mice

After 1 week of acclimatization, 18-month-old mice were given free access to TwX-containing water (or normal filtered tap water) for 1.5 months (1 month for treatment only and 0.5 months for the behavioral testing period). Body weight did not increase or differ significantly in either group during the treatment period (Figure 1A,B). Food and water intake were not significantly different between these two groups (Figure 1C,D).

### 2.2. TwX Markedly Improves Spatial Cognitive Ability in Aged Mice

To clarify the benefits of the blended supplement on learning performance, we assessed cognitive ability using the Morris water maze test. The average time-to-reach platform on the first day was similar for both groups. The arrival time of these two groups decreased day by day, and we confirmed that all mice learned the platform location (Figure 2A). The average arrival time on all trial days for the TwX-treated aged group was faster than that for the untreated aged group (statistically significant only on Day 4). The percentage of time spent on the platform by TwX-treated aged mice was not significantly different during the first 2 days, but after Day 2, it gradually became clearly (nominally) longer than in untreated aged mice (Figure 2B). The average swimming speed of the TwX-treated group (on Day 5) was not significantly different between these two groups (Figure 2C).

### 2.3. TwX Does Not Improve Coordination Skills in Aged Mice

To further characterize the benefits of the blended supplement, we measured mouse coordination using a rota-rod apparatus. No significant differences in speed-to-fall or time-to-fall scores were observed between the groups (Figure 3).

### 2.4. TwX Significantly Improves Short-Term Memory without Affecting Exploratory Behavior in the Y-Maze

Exploratory behavior and short-term memory were measured using the Y-maze task. There were no significant differences in the total number of arm entries (Figure 4, left). The alternation ratio of TwM-treated aged mice was significantly higher than that of untreated aged mice (Figure 4, right).

### 2.5. Effect of TwX Treatment on Running Distance of Aged Mice

The purpose was to examine whether the blended supplement is effective in preventing changes in leg strength caused by rearing in narrow spaces. We compared running distance before and 1 month after the treatment of aged mice with TwX or filtered tap water (Figure 5). Prior to the start of the feeding experiment, the treadmill running distance endurance of the mice was assessed, and no difference between the groups was noted (Figure 5, blue circles vs. red circles). Upon comparing the running distance in each group, there were no significant differences before and after 1 month of treatment (blue/red circles vs. blue/red squares; Figure 5A). However, when comparing the rate of increase in running distance between these two groups 1 month after administration, the score of the TwX-treated mice was significantly higher than that of the normal-aged mice (blue bars vs. red bars) (Figure 5B). 

### 2.6. Effect of TwX Treatment on the Training Effect of Exercise in Aged Mice

At the end of the 1-month treatment period, the endurance of all aged mice was measured using a treadmill test. The training was conducted three times per week, and the mice ran for 60 min at a given treadmill speed (details are described in the Materials and Methods). Two weeks after the training, all mice were subjected to the endurance test again. Both groups of aged mice showed significantly increased running distance compared with that before the start of training (Figure 6). However, treatment with TwX did not significantly enhance the effect of training.

### 2.7. TwX Does Not Alter the Secretion of Neurotrophic Factor in the Brain of Aged Mice

Treatment with the blended supplement improved spatial learning memory and exploratory behavior in aged mice. To clarify the possible mechanism underlying this effect, we measured cortex levels of neurotrophic factors such as BDNF and NGF and their cognate receptors (TrkB and TrkA, respectively) by Western blotting (Figure 7). Contrary to expectations, there were no significant differences in any of the indices.

### 2.8. Changes in Serum Parameters in Aged Mice following TwX Treatment

After all behavioral tests were completed, blood was collected, the serum was separated, and various parameters were measured (Figure 8). The levels of aspartate aminotransferase (AST), alanine aminotransferase (ALT), and total cholesterol (T-CHO) were significantly higher in TwX-treated aged mice compared with age-matched controls. There were no significant differences in other serum parameters between the presence or absence of TwX. 

## 3. Discussion

### 3.1. Treating Aged Mice with TwX Does Not Affect Normal Breeding

In this experiment, TwX, which has the same ingredients as commercially available Oxycut^®^, was added to water consumed by aged mice to examine the effects of blended antioxidants. The composition ratios of the eight active ingredients contained in TwX [15,20] and the doses administered to mice are explained in detail in the method’s section. Some of the components contained in TwX exhibit anti-obesity effects when administered alone. For example, coenzyme Q10 is an antioxidant that plays an important role in ATP synthesis in the mitochondrial respiratory chain [21]. It is also known to suppress weight gain by activating uncoupling protein 1 on the mitochondrial inner membrane in adipose tissue [22]. Although the formulation used in the study contained 1.5% coenzyme Q10, there was no change in the body weight of mice even after continuous intake for 1 month (Figure 1A). Although the weight gain rate graph appears to indicate a slight decrease, the difference was not significantly different (Figure 1B). Long-term administration to obese model mice or aged mice may have anti-obesity effects. Additionally, the TwX solution itself contains niacin, riboflavin, succinic acid, and ascorbic acid, which gives it a yellow–green color. The TwX used in this experiment was a solution, but products containing similar ingredients are sold as tablets, so one of the authors (FK) sampled the ingredients personally at the beginning of the experiment. The taste was slightly sour and bitter. We were, therefore, concerned whether the mice would drink the same amount of water as the control group; however, there was no significant difference in average water intake between groups (Figure 1D). There was also no difference in food intake during the treatment period, indicating that TwX does not affect normal food and water intake. Conversely, these results showed that, compared with control mice, the TwX-treated mice received additional vitamins and amino acids, but this did not affect body weight. In the case of vitamins, overdosing can be a problem. However, in this experiment, it can be said that continuous administration of TwX had no adverse effect on the mice.

### 3.2. Continuous Administration of TwX Improves Spatial Cognition and Exploratory Behavior in Aged Mice

In mice, cognitive function gradually declines with age and increased oxidative stress [9]. We previously confirmed that aged mice and mice with long-term vitamin E deficiency exhibit significantly reduced cognitive function and accelerated brain oxidation [20,23], and we confirmed that adding antioxidants at this time significantly improves cognitive function [24]. In the present study, we administered TwX to aged mice and conducted various behavioral tests. In the Morris water maze test, the TwX group reached the goal faster than the control group on all trial days (Figure 2A). There was no significant difference in swimming speed between these two groups. This indicates that continuous intake of TwX affects cognitive function. In the case of models of advanced age, it is thought that some degree of oxidation has already occurred in the brain, as evidenced by the accumulation of lipid peroxides, changes in fatty acid composition, and accumulation of carbonyl proteins [25]. The present results suggest that the intake of antioxidants, even after certain levels of oxidative substances have accumulated, may affect cognitive function. However, this experiment did not actually measure the oxidative state of the brain, as evidenced by lipid peroxides. Kusaki et al. reported that treatment with TwX for 2 weeks in an acute ischemic mouse model reduced the levels of 8-hydroxy-2′-deoxyguanosine and 4-hydroxy-2-nonenal by immunohistochemical analysis [19]. There are many reports on the neuroprotective effects of vitamin administration via antioxidant function, including our previous studies [24,26]. However, most of them involve administering fat-soluble vitamins or a mix of fat-soluble and water-soluble vitamins. We thought that most of these effects were fat-soluble, but recently, there have been some papers that show the neuroprotective effects of water-soluble vitamins such as vitamins B and C [27,28]. This may be related to how easily they are absorbed, how well they reach the brain, and how long they remain in the blood. Amino acids such as fumaric acid and L-glutamine, which are included in TwX, may be involved. TwX is primarily a water-soluble vitamin. The detailed mechanism behind this result is unknown. It is unclear whether this result affects an antioxidant effect, a reduction in the degree of oxidation in the brain, or other effects. Further investigation is needed to clarify this point. 

There was no significant difference in fall time between these two groups in the rota-rod test (Figure 3). This test is known to have a significant correlation with body weight [29]. In this experiment, there was no significant difference in body weight following TwX administration, so it is thought that there was also no difference in the rota-rod test results.

The Y-maze test is an indicator of short-term memory and exploratory behavior [30]. In this experiment, there was no significant difference in the number of arm intrusions between these two groups, suggesting that exploratory behavior was not affected by the presence or absence of TwX (Figure 4). In contrast, the spontaneous alternation rate was significantly increased by TwX administration compared with the no TwX treatment group. This result indicates that TwX may improve short-term memory. Combined with the water maze test results, this result suggests that TwX contributes to the maintenance and improvement of spatial task performance ability. There are reports that cognitive function is improved by ingestion of the vitamin B, vitamin C, and coenzyme Q10 contained in TwX [31,32,33]. Travica et al. found a strong correlation between plasma vitamin C levels and working memory [32]. Monsef et al. reported that the administration of coenzyme Q10 to middle-aged male Wistar rats for 45 days significantly improved cognitive function [33].

It is unclear why adding different vitamins similarly improves cognitive function. However, if ‘antioxidant’ is the keyword, all of these effects may be the result of antioxidant activity. Vitamins C and E are more effective when administered in combination [34]. Because TwX contains multiple active ingredients, it can be highly effective. This type and ratio may be good, but we have not compared other ratios, so we will need to consider other combinations in the future. As a significant effect was observed in behavioral tests, we hypothesized that the secretion of neurotrophic factors in the brain was activated, but there were no significant differences in the levels of secreted NGF or BDNF between these two groups (Figure 7). Although the detailed mechanism of action is unknown, further investigation is necessary because it is thought to contribute to membrane stability in brain neurons, affect synaptic plasticity, and enhance hormone secretion.

### 3.3. TwX Treatment Does Not Enhance Exercise Training Capacity but May Prevent Muscle Weakness

As humans age, muscle strength decreases, and when this worsens, frailty and sarcopenia can develop [35]. Frailty and sarcopenia cause declines in muscle strength and cognitive function, making them risk factors for severe age-related neurodegenerative diseases [36]. Therefore, maintaining muscle strength provides an opportunity for action and is very important [37]. As laboratory animals, mice are usually kept in cages with limited space for activity. Therefore, we conducted an endurance test using a treadmill before and after administering TwX to examine potential differences in running distance. There was no significant difference in running distance between these two groups before and after treatment with water alone or water containing TwX (Figure 5A). However, when comparing the rate of increase in running distance between the groups at the end of the treatment period, the rate of increase in running distance was significantly higher in the TwX treatment group (Figure 5B). 

To enhance the training effect, mice were subjected to exercise training, and TwX was administered for an additional 2 weeks after the initial TwX administration period. Unfortunately, although training effects were observed in both groups, there were no significant differences between the groups (Figure 6). Finally, we analyzed a multivariate approach using multiple regression. Treatment with TwX tended to result in higher scores for all, but there were no significant differences between these two groups. The mice used in this experiment are already old. It is possible that muscle mass had already declined before TwX administration, and it is also possible that administration of TwX might have enhanced muscle strength improvement even if the animals had been kept in a closed space and not been trained. 

Several studies have examined the relationship between insufficient vitamin intake and the risk of frailty. Balboa-Catillo et al. reported that lower dietary intakes of vitamins B6, C, E, and folic acid are associated with a higher risk of frailty [38]. It was also reported that people with high physical activity and strong muscles have higher levels of coenzyme Q10 in their blood [39]. These results indicate that although continuous intake of TwX does not increase muscle strength, it may be effective in treating muscle weakness associated with a lack of exercise and aging. Serological tests showed significant improvements in ALT, AST, and T-CHO levels, suggesting improved liver function (Figure 8). We are currently practicing harvesting various types of muscle and fat. In the future, we wish to perform Western blotting and PCR on each tissue to elucidate the mechanism of muscle strength enhancement by TwX administration. It is well known that vitamin E is effective in treating non-alcoholic fatty liver disease [40]. Considering this result alone, although it does not have an anti-obesity effect, TwX may be beneficial because oxidation is deeply involved in the development of fatty liver caused by obesity [41].

## 4. Materials and Methods

### 4.1. Animals

All animal experiments were approved by the Animal Protection and Ethics Committee of Shibaura Institute of Technology (Approval Number: #21005; Approval Date: 26 August 2021). Eighteen-month-old C57BL/6 mice were gifted by our collaborator, the Division of Anti-oxidant Research, Life Science Center, Gifu University (Gifu, Japan). After 1 week of habituation, the mice were assigned to two groups, which were provided with either TwX-containing drinking water or normal filtered tap water ad libitum. All mice were provided a control diet (Labo MR Stock, Nosan Corp., Kanagawa, Japan) ad libitum. The number of experimental animals in each group is indicated in the figure captions.

After 1 month of consuming the experimental drink, the mice were anesthetized by inhalation with isoflurane and euthanized by decapitation. The vaporization concentration of isoflurane was 3 to 5%, and anesthesia was administrated in a fume hood chamber. After euthanasia, the brain was removed while cooling and divided into the cerebral cortex and the rest, and blood was also collected for serum test. During the experimental period, mice were maintained under controlled temperature (22 ± 2 °C) conditions and a 12-h/12-h light/dark cycle. Food and water intake and body weight were measured weekly. All chemical reagents other than all feed, water, and TwX were obtained from FUJIFILM Wako Pure Chemical Corp. (Osaka, Japan).

### 4.2. Preparation of Blended Antioxidant Supplement Solution

The blended antioxidant supplement (formula name: Twendee X) was produced by the Division of Antioxidant Research, Gifu University [20], and developed by TIMA (Balzers, Liechtenstein). TwX is also sold in a similar formulation by TIMA Tokyo Inc. (Tokyo, Japan) under the product name Oxycut; however, it differs from the formulation used in this experiment. The sample of TwX used in this study was not a commercially available product but instead produced by Gifu University. Commercially available Oxycut is a tablet, but the sample used in this study was a liquid ampoule. TwX comprises 8 substances, including coenzyme Q10 (1.5%), ascorbic acid (35.5%), L-glutamine (32.0%), L-cysteine (18.9%), niacin (0.75%), succinic acid (3.8%), fumaric acid (3.8%), and riboflavin (1.5%). The mice were fed TwX (20 mg/kg/day) via oral intake from 18 months of age until euthanasia. Beverages were prepared by taking into account the average body weight and average water intake of mice at 18 months of age.

### 4.3. Behavioral Assessment

#### 4.3.1. Morris Water Maze

Spatial learning ability was assessed using a Morris water maze apparatus for mice [13,42]. The maze apparatus constructed of acrylic resin (120 cm in diameter and 30 cm in height; #MWM-04M, Muromachi Kikai Co., Ltd., Tokyo, Japan) was used for the experiments. To measure staying time in the platform area, the bottom of the pool was divided into four sections with white tape. Four markers were placed around the wall. An escape platform was placed in the center of one quadrant. The pool water temperature was maintained at around 22 °C. To become familiar with the water maze apparatus, mice were allowed to swim freely for 60 s without a platform for 3 days before the start of the experiment. The trials were conducted four times per day for 5 consecutive days. This was performed every 3 h at the same time each day (starting at 1000, 1300, 1600, and 1900 h). Goal time, swimming distance, speed, and the ratio of staying time in the quadrant containing the platform were measured using ANY-maze software (ver. 4.98; Stoelting Co., Wood Dale, IL, USA).

#### 4.3.2. Rota-Rod Test

The rota-rod (#MK-670, Muromachi Kikai Co., Ltd.) test was used to assess coordinated movement ability [29]. The rod speed was 5 rpm for the first 60 s, accelerated to 50 rpm for 60 s, and then held at a constant speed. The time-to-fall and the rotational speed of the rod at the time of fall were measured.

#### 4.3.3. Y-Maze Test

Short-term memory was assessed using a Y-maze apparatus (Muromachi Kikai Co., Ltd.) [30]. The mouse was allowed to move freely in the maze for 10 min, and after collecting a video, ANY-maze software was used to calculate the alternation behavior rate.

### 4.4. Treadmill Exercise Experiment

#### 4.4.1. Endurance Test

The endurance of each mouse was assessed using a treadmill (#MK-690; Muromachi Kikai Co., Ltd., Tokyo, Japan) [9]. Prior to the start of the experiment, the mice were allowed to run or walk freely for 10 min (the belt was not active during this phase) to acclimate to the treadmill. Next, the machine was set to accelerate 1 m/min at an incline of 0°; to habituate the mice to running, the treadmill was accelerated from 1 to 10 m/min every 2 min for 20 min. After a 10-min interval, the treadmill was set at an incline of 10° and accelerated from 10 to 30 m/min at a rate of 2 m/min every 4 min, and the mice were allowed to continue running until exhausted. The session ended when the mouse touched the shock grid for a total of 5 s. The time-to-exhaustion and running distance were measured for each mouse. The difference in endurance before and after TwX treatment was evaluated based on running distance. In this experiment, mice were not subjected to training, and the mice were housed in the animal cages for 2 months.

#### 4.4.2. Treadmill Training

Two months after administration of each experimental water and completion of the endurance tests, each group of mice was evenly divided into two subgroups based on running distance. One group trained on the treadmill three times per week, with the mice running for 60 min at a 0° incline. The running speed was 15 m/min in the first week and 18 m/min in the second week. The second subgroup was housed normally in individual cages without training for the same period as the first group.

#### 4.4.3. Western Blotting

The cerebral cortex was homogenized using radioimmunoprecipitation assay buffer [9]. Protein content was determined using a kit (Bio-Rad protein assay, #500-0006JA, Bio-Rad Laboratories, Inc., Hercules, CA, USA). Twenty micrograms of each protein extract were separated by electrophoresis and transferred to a membrane (CelarTrans, 0.2 μm, #030-25643, FUJIFILM Wako Pure Chemical Corp.). All gels used 12% sodium dodecyl sulfate-polyacrylamide. The membranes were blocked with a blocking solution (2% non-fat skim milk in Tris-HCl-buffered saline [TBS-T, pH 7.6] containing 0.1% Tween 20) for 1 h at room temperature (R/T). The membranes were treated overnight at 4 °C with the following primary antibodies: anti-brain-derived neurotropic factor (BDNF) (N-20) rabbit polyclonal antibody, 1:2500 (#ab-108319, Abcam plc, Cambridge, UK), anti-nerve growth factor (NGF) (H-20) rabbit polyclonal antibody, 1:4000 (#sc-548, Santa Cruz Biotechnology (SCBT), Inc., Dallas, TX, USA), anti-tropomyosin receptor kinase A (TrkA) (763) rabbit polyclonal antibody, 1:4000 (#sc-118, SCBT), and anti-TrkB (H-181) rabbit polyclonal antibody, 1:250 (#sc-8316, SCBT). Horseradish peroxidase-conjugated anti-rabbit IgG antibody (Promega Corp., Madison, WI, USA) was used as a secondary antibody at 1:4000 dilution for 1 h at R/T. The bands were made chemiluminescent to measure the band intensities using a specialized detection reagent (Immobilon; Merck KGaA, Darmstadt, Germany). The relative band intensities were determined using a LAS-3000 (FUJIFILM Corp., Tokyo, Japan). The expression ratios were calculated using Image J software (1.52a; National Institutes of Health, Bethesda, MD, USA).

#### 4.4.4. Serum Parameters

Total protein (TP), albumin, aspartate aminotransferase (AST), alanine aminotransferase (ALT), lactic acid dehydrogenase, total cholesterol (T-CHO), triglycerides, and high-density lipoprotein were measured by an external vendor (Oriental Yeast Co., Ltd., Tokyo, Japan).

#### 4.4.5. Statistical Analysis

Data are expressed as mean ± standard error (SE), and differences were analyzed using Prism 9.2.0 (GraphPad Software, San Diego, CA, USA). Differences were considered significant at *p* < 0.05. Detailed statistical methods are described in the individual figure captions.

## 5. Conclusions

Long-term intake of blended antioxidant supplements may be effective, even considering the effects of aging and related increased oxidation in the body. In this study, spatial learning ability and short-term memory were significantly improved in blended supplement-treated aged mice. The detailed mechanism could not be confirmed. There are also reports that depression and anxiety behaviors are improved by taking vitamins, so there may be other effects other than antioxidants [43]. Frailty and sarcopenia are now serious problems and risk factors for dementia. Although the mechanism is unknown, it is groundbreaking that taking supplements may be able to prevent muscle weakness. Taking blended supplements over a long period of time may contribute to improving quality of life.

## Figures and Tables

**Figure 1 ijms-25-02804-f001:**
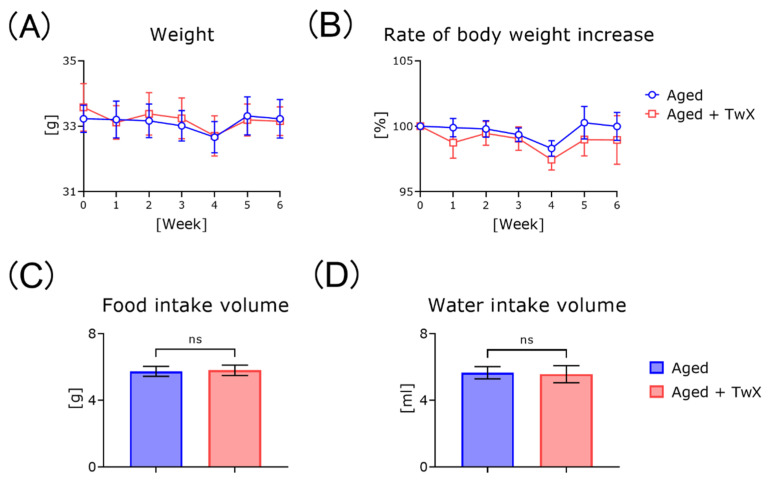
Changes in various measurement indices during breeding in TwX-treated aged mice. Body weights of the mice from 18 to 19.5 months old were shown in (**A**). Food and water intake for each mouse group during the final week (**C**,**D**). Aged, n = 8; Twendee X treated aged mice (Aged + TwX), n = 8. All data were statistically analyzed using a two-way analysis of variance (**A**,**B**). The food and water intake for the final week were analyzed using a two-tailed, non-paired Student’s *t*-test (**C**,**D**).

**Figure 2 ijms-25-02804-f002:**
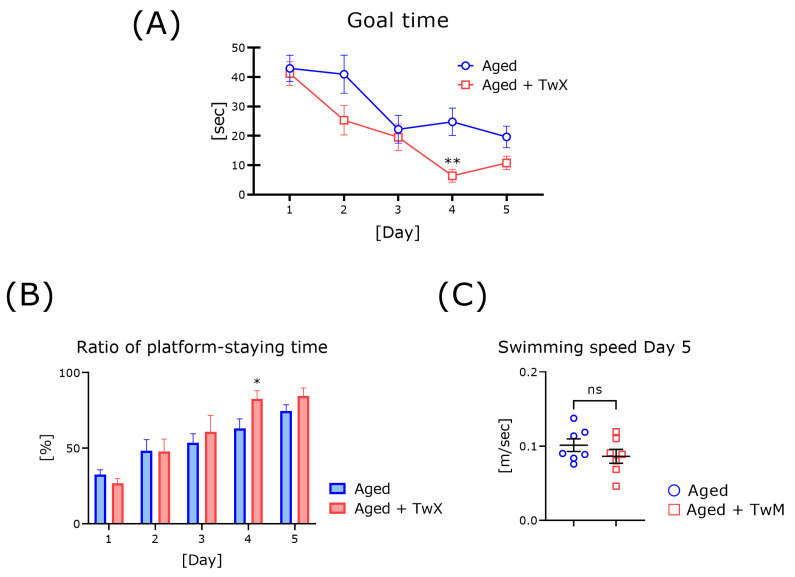
TwX improves spatial learning ability in aged mice. The average goal time (escape latency) in the spatial cognition test is shown in (**A**). The percentage of time spent in the quadrant where the platform was located is shown in (**B**). The average swimming speed on Day 5 of the experiment is shown in (**C**). Twendee X-treated aged mice (Aged + TwM; n = 7) and age-matched control mice (Aged; n = 7) were used. * *p* < 0.05, ** *p* < 0.01, ns means no significance, vs. the age-matched control group. These data are shown as mean ± SE. Data for time-to-goal were statistically evaluated using a two-way analysis of variance (**A**). Movement speed (swimming) and the ratio of staying time in the platform quadrant were statistically evaluated using the two-tailed, non-paired Student’s *t*-test (**B**,**C**).

**Figure 3 ijms-25-02804-f003:**
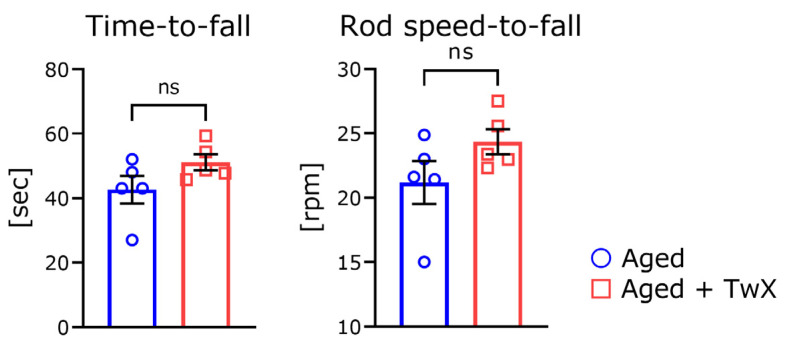
Time-to-fall and rod speed-to-fall in the rota-rod test. The rod speed-to-fall and time-to-fall are shown in the left and right graphs, respectively. Aged mice (Aged), n = 6; Twendee X-treated aged mice (Aged + TwX), n = 6. ns means no significance. Data are shown as mean ± SE. These data were statistically analyzed using the two-tailed, non-paired Student’s *t*-test.

**Figure 4 ijms-25-02804-f004:**
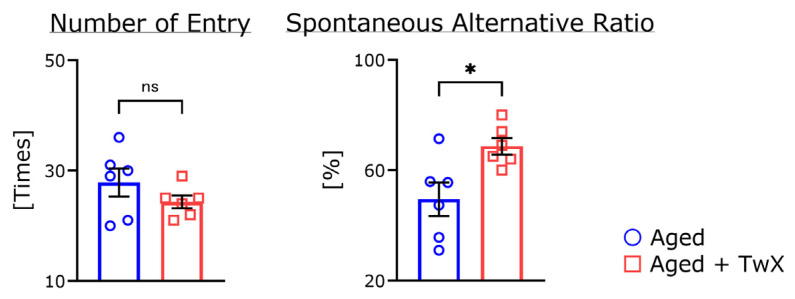
Alternation score in the Y-maze test. The total number of entries and spontaneous alternation ratio are shown in the left and right graphs, respectively. Aged mice (Aged), n = 6; Twendee X-treated aged mice (Aged + TwX), n = 6. * *p* < 0.05, ns means no significance. These data are shown as mean ± SE. These data were statistically analyzed using the two-tailed, non-paired Student’s *t*-test.

**Figure 5 ijms-25-02804-f005:**
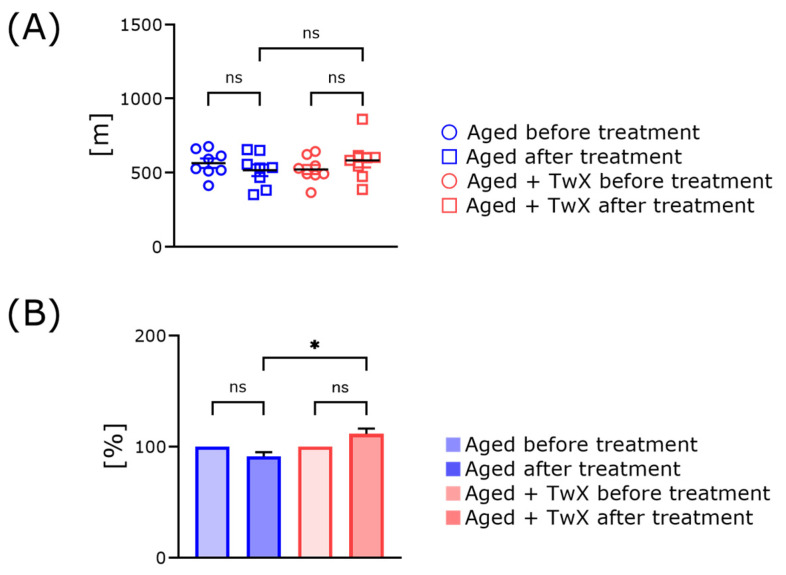
Change in treadmill running distance of aged mice before and after TwX treatment. Circles indicate the running distance before starting treatment with TwX or filtered tap water, and squares indicate the running distance 1 month after treatment with each drinking water (**A**). Relative running distance (**B**). Before starting treatment with each water, the running distance was set to 100% (**B**). Aged mice (Aged), n = 8; Twendee X-treated aged mice (Aged + TwX), n = 8. * *p* < 0.05. ns means no significance. The data are shown as mean ± SE. Data were statistically evaluated using a one-way analysis of variance followed by the Tukey–Kramer test.

**Figure 6 ijms-25-02804-f006:**
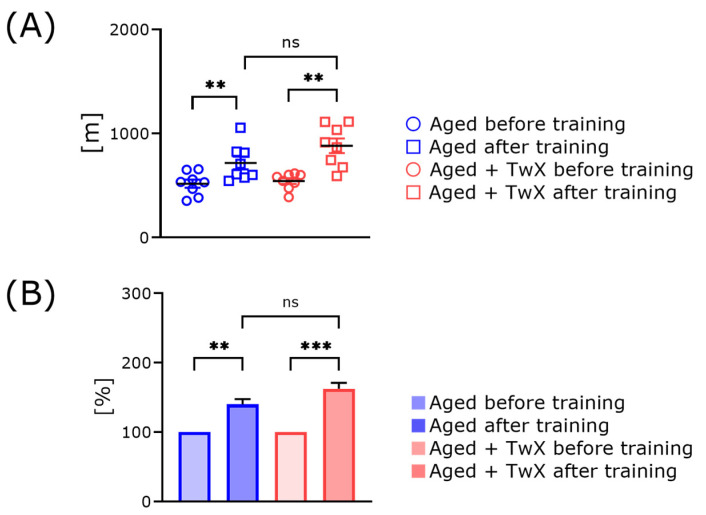
Change in treadmill running distance of aged mice with or without TwX treatment before and after 2 weeks of training. Circles indicate the running distance before the start of training, and squares indicate the running distance 2 weeks after training (**A**). Relative running distance (**B**). Before starting the treatment with each water, the running distance was set to 100% (**B**). Aged (n = 8), Aged + TwX (n = 8). ** *p* < 0.01, *** *p* < 0.001, ns means no significance. The data are shown as mean ± SE. Data were analyzed using a one-way analysis of variance followed by the Tukey–Kramer test.

**Figure 7 ijms-25-02804-f007:**
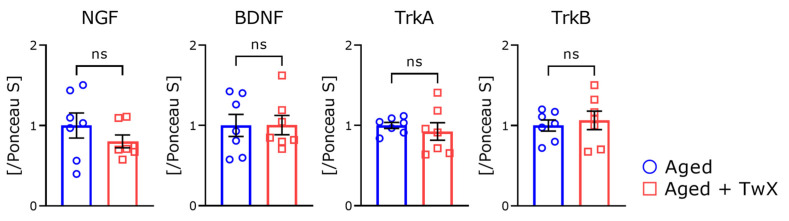
Changes in the levels of neurotrophic factor-related protein expressions in the brains of aged mice with or without TwX treatment. All Western blotting analyses were performed using the cerebral cortex region. The relative band intensity ratio of all proteins to the intensity of Ponceau S stained bands is shown, with ratios of normal-aged samples set to 1. Aged mice (Aged), n = 7; Twendee X-treated aged mice (Aged + TwX) n = 7, ns means no significance. The data are shown as mean ± SE. Comparisons were performed using the two-tailed, non-paired Student’s *t*-test.

**Figure 8 ijms-25-02804-f008:**
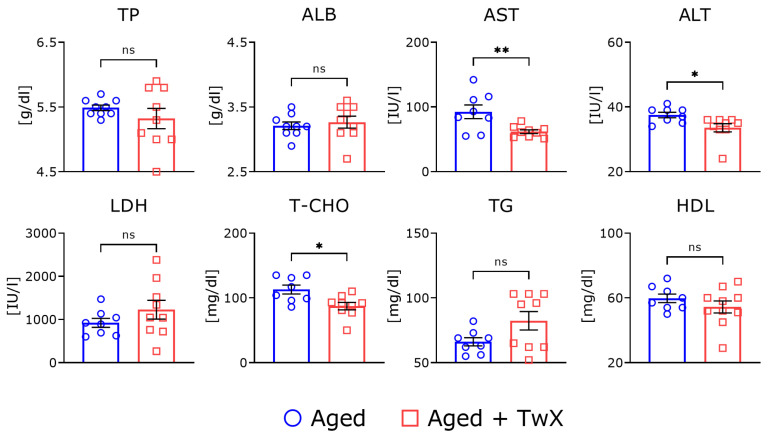
Serum parameters in aged and TwX-treated aged mice. Aged mice (Aged), n = 7; Twendee X treated aged mice (Aged + TwX), n = 7. These data are shown as mean ± SE. * *p* < 0.05, ** *p* < 0.01, ns means no significance. Comparisons were performed using the two-tailed, non-paired Student’s *t*-test. TP, total protein; ALB, albumin; AST, aspartate aminotransferase; ALT, alanine aminotransferase; LDH, lactic acid dehydrogenase; T-CHO, total cholesterol; TG, triglycerides; HDL-C, high-density lipoprotein cholesterol.

## Data Availability

All data generated or analyzed during this study are included in this published article.

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
