# Peer review of "A Blended Vitamin Supplement Improves Spatial Cognitive and Short-Term Memory in Aged Mice"

_ijms, 2024, doi:10.3390/ijms25052804_

Round 1
Reviewer 1 Report
Comments and Suggestions for Authors
Thank you for the opportunity to review the article.
Article by Fukui et al. entitled “A blended vitamin supplement improves spatial cognitive and short-term memory in aged mice” presents the various effects of continuous administration of a blended supplement to aging mice. I read this article with interest. However, before publication, a few things need to be corrected in the text.
Abstract: Authors should complete 1-2 sentences about the methods used. A p-value should be added where a statistically significant difference is described.
Introduction:
Sentence: “Therefore, to prevent or delay the various functional declines associated with aging, it is important to continually prevent oxidation in the body.” needs reference.
Abbreviations such as SOD, CAT, and GSHPx should be expanded upon first use so that a reader unfamiliar with the topic of oxidative stress can easily understand the topic of the article.
Sentences: “This may be due to a variety of factors, including weakening of in vivo antioxidant defense mechanisms due to aging and exposure to external stress. Therefore, it is difficult to maintain the redox balance associated with aging simply by ingesting antioxidants from the diet.” need references.
Sentence: “A wide variety of supplements are available, and anyone can easily purchase them.” should be remove - this sentence is written in too colloquial language.
The introduction should be enriched with the topic of the impact of vitamin supplementation on spatial cognitive and short-term memory. Adequate rationale in Introduction section also should be added.
Material and methods:
“The brain (cerebral cortex) and blood (for serum) were collected for corresponding biochemical analyses” – authors should complete the description of these methods – this description is too brief. Methods should be described clearly and in sufficient detail to enable reproducibility.
The remaining methods are described sufficiently.
Results:
Font size should be standardized on figures.
The abbreviations used, e.g. AST, ALT and T-CHO, should be explained upon first use.
The results are well presented and well described. The quality of the figures and their readability is correct.
Discussion:
Discussions should be more critical and comprehensive, including fully address study limitations.
Conclusion:
The conclusions should be rewritten - they are too general.
References:
The references are well selected and up to date.
Author Response
Article by Fukui et al. entitled “A blended vitamin supplement improves spatial cognitive and short-term memory in aged mice” presents the various effects of continuous administration of a blended supplement to aging mice. I read this article with interest. However, before publication, a few things need to be corrected in the text.
Abstract: Authors should complete 1-2 sentences about the methods used. A p-value should be added where a statistically significant difference is described.
[Answer] Thank you for your comment, I added extra information of experimental results.
Introduction:
Sentence: “Therefore, to prevent or delay the various functional declines associated with aging, it is important to continually prevent oxidation in the body.” needs reference.
[Answer] Thank you very much for your comment. I added reference.
Abbreviations such as SOD, CAT, and GSHPx should be expanded upon first use so that a reader unfamiliar with the topic of oxidative stress can easily understand the topic of the article.
[Answer] Thank you very much for your comment. I corrected these words.
Sentences: “This may be due to a variety of factors, including weakening of in vivo antioxidant defense mechanisms due to aging and exposure to external stress. Therefore, it is difficult to maintain the redox balance associated with aging simply by ingesting antioxidants from the diet.” need references.
[Answer] Thank you very much for your comment. I added reference.
Sentence: “A wide variety of supplements are available, and anyone can easily purchase them.” should be remove - this sentence is written in too colloquial language.
[Answer] Thank you very much for your comment. I deleted this sentence.
The introduction should be enriched with the topic of the impact of vitamin supplementation on spatial cognitive and short-term memory. Adequate rationale in Introduction section also should be added
[Answer] Thank you very much for your comment. I added sentence and references about TwX.
Material and methods:
“The brain (cerebral cortex) and blood (for serum) were collected for corresponding biochemical analyses” – authors should complete the description of these methods – this description is too brief. Methods should be described clearly and in sufficient detail to enable reproducibility.
[Answer] Thank you very much for your comments. I added extra information of making tissue samples from the mice.
The remaining methods are described sufficiently.
[Answer] Thank you very much for your comment.
Results:
Font size should be standardized on figures.
[Answer] Although everything is unified in the diagram, the font size will be different because it will be enlarged or reduced when offset.
The abbreviations used, e.g. AST, ALT and T-CHO, should be explained upon first use.
[Answer] Thank you very much for your comment. I corrected them.
The results are well presented and well described. The quality of the figures and their readability is correct.
[Answer] Thank you very much for your comment.
Discussion:
Discussions should be more critical and comprehensive, including fully address study limitations.
[Answer] I added sentences and references.
Conclusion:
The conclusions should be rewritten - they are too general.
[Answer] Most of the conclusions have been rewritten.
References:
The references are well selected and up to date.
[Answer] Thank you very much
Reviewer 2 Report
Comments and Suggestions for Authors
The paper entitled ” A blended vitamin supplement improves spatial cognitive and short-term memory in aged mice” is an interesting work. The authors describe the effects of dietary integration with TwX in old mice. I find some issues that must be analysed by authors:
1. Introduction: this is too concise and does not describe previous works that have characterized the effects of single components of TwX.
2. Methods: In the statistical analysis the post hoc tests must be mentioned.
3. Results: The figures lack accuracy (for example remove ticks in the x-axes, or describe the unit of measure for time in the y-axes
4. Results: please analyse all the behavioural scores together using a multivariate approach such as PCA.
5. Discussion and conclusion: these sections need an improvement to limit speculations and support your data with previously published data provided by a single component of your supplementation.
Author Response
The paper entitled ” A blended vitamin supplement improves spatial cognitive and short-term memory in aged mice” is an interesting work. The authors describe the effects of dietary integration with TwX in old mice. I find some issues that must be analysed by authors:
- Introduction: this is too concise and does not describe previous works that have characterized the effects of single components of TwX.
[Answer] Thank you for your valuable comment. I added sentences about TwX and references.
- Methods: In the statistical analysis the post hoc tests must be mentioned.
[Answer] Thank you very much for your valuable feedback. For two groups, a t-test was used. For groups of three or more, Tukey was performed after the ANOVA test. Therefore, all data have already been analyzed by post-hoc analysis.
- Results: The figures lack accuracy (for example remove ticks in the x-axes, or describe the unit of measure for time in the y-axes
[Answer] Thank you very much for your comment. I checked and corrected all figures.
- Results: please analyse all the behavioural scores together using a multivariate approach such as PCA.
[Answer] Thank you for your comment. Following your comment, I analyzed the multivariate approach. However, there was no significant difference between the two groups. Added sentence to discussion section. (Please check attachment file)
- Discussion and conclusion: these sections need an improvement to limit speculations and support your data with previously published data provided by a single component of your supplementation.
[Answer] Thank you very much for your comment. Following your comment, I added extra sentences and references.

Round 2
Reviewer 1 Report
Comments and Suggestions for Authors
The authors have significantly improved the article. The manuscript may be published.
Reviewer 2 Report
Comments and Suggestions for Authors
Dear authors,
I appreciated the new version of this article.